# Suramin Disturbs the Association of the N-Terminal Domain of SARS-CoV-2 Nucleocapsid Protein with RNA

**DOI:** 10.3390/molecules28062534

**Published:** 2023-03-10

**Authors:** Chenyun Guo, Hao Xu, Xiao Li, Jiaxin Yu, Donghai Lin

**Affiliations:** Key Laboratory of Chemical Biology of Fujian Province, Department of Chemical Biology, College of Chemistry and Chemical Engineering, Xiamen University, Xiamen 361005, China

**Keywords:** suramin, SARS-CoV-2, N-NTD, NMR, protein interaction

## Abstract

Suramin was originally used as an antiparasitic drug in clinics. Here, we demonstrate that suramin can bind to the N-terminal domain of SARS-CoV-2 nucleocapsid protein (N-NTD) and disturb its interaction with RNA. The BLI experiments showed that N-NTD interacts suramin with a dissociate constant (K_d_ = 2.74 μM) stronger than that of N-NTD with ssRNA-16 (K_d_ = 8.37 μM). Furthermore, both NMR titration experiments and molecular docking analysis suggested that suramin mainly binds to the positively charged cavity between the finger and the palm subdomains of N-NTD, and residues R88, R92, R93, I94, R95, K102 and A156 are crucial for N-NTD capturing suramin. Besides, NMR dynamics experiments showed that suramin-bound N-NTD adopts a more rigid structure, and the loop between β2-β3 exhibits fast motion on the ps-ns timescale, potentially facilitating suramin binding. Our findings not only reveal the molecular basis of suramin disturbing the association of SARS-CoV-2 N-NTD with RNA but also provide valuable structural information for the development of drugs against SARS-CoV-2.

## 1. Introduction

SARS-CoV-2 is the pathogen of Corona Virus Disease 2019 (COVID-19), consisting mainly of four known structural proteins, including Spike (S), Membrane (M), Envelope glycoproteins (E) and Nucleocapsid (N) [1]. Among them, the N protein is a highly conserved protein, playing essential roles in binding viral RNA and packing it into the ribonucleoprotein (RNP) complex [2,3,4,5,6]. The N protein is also implicated in host cell metabolism; regulating biological activities, such as cell cycle progression; host-pathogen interaction and cell pyroptosis [7,8,9,10,11,12]. Besides, the N protein can suppress the antiviral immunity of the host cell by disturbing the RIG-I-like receptor pathway [13]. Therefore, the N protein is a promising drug target for treating COVID-19 [14,15,16].

The sequence similarity of the N protein between SARS-CoV-2 and SARS-CoV is as high as 90%, indicating that both proteins share a similar structural pattern [17]. The SARS-CoV-2 N protein contains two conserved domains (N-NTD: 44–174 a.a.; N-CTD: 255–363 a.a.) and three intrinsically disordered regions (IDRs: 1–43 a.a., 175–254 a.a., 364–419 a.a.) [15]. The three-dimensional (3D) structures of N-NTD and N-CTD domains have been determined [18,19,20,21]. Tatsuhito et al. found that the N-NTD and N-CTD structures of SARS-CoV-2 resemble those of other β-coronaviruses, such as SARS-CoV and MERS-CoV [22]. SARS-CoV-2 N-NTD shows a right-handed fist shape composed of a palm subdomain and a basic finger subdomain. The palm subdomain is the core structure of N-NTD consisting of one five-stranded β-sheet with the topology β4-β2-β3-β1-β5 and two short flanking α-helices. The basic finger subdomain is a protruding β-hairpin (86–109 a.a.) between β2 and β3, mostly containing basic residues. The positively charged canyon between the finger and the palm subdomains acts as the main RNA-binding site, and residues R92, R107 and R149 play key roles in capturing RNA [18,19]. In addition, it was reported that N-NTD had different affinities for binding different RNA fragments, depending on the length and conformation of the tested RNA fragments [19,23].

The emergence of the COVID-19 pandemic represents an increased risk to global public health. Effective drugs are urgently needed against the pandemic. The repurposing or redesign of existing drugs is a potential way to accelerate this process. Many antiviral, anti-inflammatory drugs were tested for the treatment of COVID-19, such as rapamycin, saracatinib, camostat, and so on [24]. Suramin has been reported to have the potency to prevent the progression of SARS-CoV-2 infection in human airway epithelial cells; however, the underlying molecular mechanisms still remain unclear [25]. Raphae et al. found that suramin and quinacrine can cooperatively inhibit the SARS-CoV-2 main protease (3CL^pro^) in vitro [26]. The inhibitory activities of these two drugs on 3CL^pro^ still need to be further verified using in vivo experiments. Recently, Yin et al. demonstrated that suramin can block the binding of RNA to the SARS-CoV-2 RNA-dependent RNA polymerase (RdRp) [27]. These results suggest that suramin has the potency to be developed as an anti-SARS-CoV-2 drug. Furthermore, suramin has been previously used to treat African trypanosomiasis and parasite infections, and it could bind to the enteroviral nucleocapsid protein and inhibit its attachment to the human host cell [28,29]. As a symmetrical compound with three sulfonic groups at each end carrying six negative charges under physiological pH conditions (Figure 1), suramin more likely binds to positively charged sites in the pockets of proteins. We thus speculated that suramin might bind to the positively charged pocket in the N-NTD domain of the SARS-CoV-2 N protein (SARS-CoV-2 N-NTD).

Herein, we determined the binding affinity of suramin to SARS-CoV-2 N-NTD and disclosed the structural basis of suramin disturbing the association of N-NTD with RNA. Our results may be beneficial to both the in-depth understanding of molecular mechanisms underlying SARS-CoV-2 viral assembly and the development of specific drugs against COVID-19.

## 2. Results and Discussion

### 2.1. Suramin Disturbs the Association of SARS-CoV-2 N-NTD with RNA

The N protein of SARS-CoV-2 can pack the RNA genome into the ribonucleoprotein (RNP) complex, which is crucial for virus replication. The NTD domain of the N protein (N-NTD) plays an important role in binding RNA. Many previous studies have demonstrated that the SARS-CoV-2 N-NTD can bind many RNA fragments with different lengths [19,23]. Here, we analyzed the interaction of SARS-CoV-2 N-NTD with the ssRNA-16 fragment (ssRNA-16: 5′-AUAUGGAAGAGCCCUA-3′) derived from the non-translated region at the 3 ‘end of SARS-CoV-2 genome [30]. The EMSA experiment showed that when SARS-CoV-2 N-NTD were mixed with ssRNA-16 at a molar ratio of 1:1, most of the RNA bound to SARS-CoV-2 N-NTD, and only a small amount of free RNA remained (Figure 2, lane 2). With the concentrations of suramin increasing, RNA was gradually dissociated from N-NTD, and the corresponding band of free RNA was gradually strengthened (Figure 2, lane 3–9). These results showed that suramin could prevent the binding of SARS-CoV-2 N-NTD with RNA in vitro. Therefore, suramin could be developed as a potential inhibitor against SARS-CoV-2.

### 2.2. Suramin Has a Higher Affinity for Binding SARS-CoV-2 N-NTD Than RNA

To quantitatively analyze the intermolecular interaction of SARS-CoV-2 N-NTD with either suramin or RNA, we first determined the binding affinity of N-NTD with ssRNA-16 using BLI experiments, obtaining a K_d_ value of 8.37 μM. It was previously reported that binding affinities of N-NTD with different RNA fragments were in the range of 6–190 μM, depending on the length and conformation of the tested RNA fragments [17]. It seemed that both the association and dissociation processes of N-NTD with suramin (Figure 3a) were slower than those with RNA (Figure 3b). Furthermore, we determined the affinity of N-NTD for binding suramin (K_d_ = 2.74 μM), which was stronger than those for binding all reported RNA fragments so far. Therefore, suramin could competitively bind to N-NTD by replacing RNA. Additionally, the affinity of suramin for binding to N-NTD was also stronger than that for binding to 3CL^pro^ (K_d_ = 59.7 μM).

### 2.3. Suramin Shares Similar Binding Areas with RNA on SARS-CoV-2 N-NTD

We explored the potential binding sites of suramin or RNA on SARS-CoV-2 N-NTD by observing the peak change in the 2D ^1^H-^15^N HSQC spectra of N-NTD before and after ligand titration. As we know each residue of N-NTD corresponds to one peak in its 2D ^1^H-^15^N HSQC spectrum, and the location and intensity of the peak are closely related to the chemical environment of its corresponding residue, therefore, once a residue is involved in ligand binding, its peak location or intensity will change. Based on the chemical shift data of the N-NTD protein deposited in the Biological Magnetic Resonance Bank (BMRB ID: 35411), we identified binding sites of either suramin or RNA on N-NTD by following the changes of peaks in the 2D ^1^H-^15^N HSQC spectra recorded during the titration experiments. Firstly, we titrated ssRNA-16 into ^15^N-labeled N-NTD protein solution and observed 11 obviously shifted peaks in the 2D ^1^H-^15^N HSQC spectra of N-NTD (Figure 4a), corresponding to residues G60, A90, G96, G97, K143, I146, A152, Q160, G170, G175 and S176. Additionally, 24 residues underwent significant peak broadening, including L56, D63, L64, K65, F66, N75, T91, R92, R93, I94, R95, K100, K102, D103, S105, T135, L139, N150, A156, L159, T165, T166, A173 and R177. These peak perturbations suggested that N-NTD possessed a medium affinity (~μM) to RNA, which was consistent with the results obtained from the above-described BLI experiments. Further, these perturbed residues were identified as potential binding sites of ssRNA-16 on N-NTD. Then we mapped these significantly perturbed residues onto the 3D structure of N-NTD (PDB ID: 6YI3, Figure 4b), and found they formed a U-shaped binding region between the finger and the palm subdomains.

On the other hand, 41 peaks of N-NTD showed obvious shift or/and broaden during the suramin titration (Figure 4c), indicating a moderate interaction between suramin and N-NTD with a medium affinity (~μM), which is also consistent with the results from the BLI experiments. A total of 11 residues were obviously shifted, including G44, N48, S51, R68, R95, D128, G129, T135, V158, L167 and G179, and 30 residues were broaden covering N47, W52, T54, L56, T57, H59, G60, K61, L64, K65, A90, T91, R93, I94, G96, G99, K100, M101, K102, R107, W108, L139, R149, A155, A156, L159, Q160, Q163, Y172 and G175. These 41 perturbed residues are mapped on the 3D structure of N-NTD (PDB ID: 6YI3, Figure 4d), which are mostly located at the junction between the finger and the palm subdomains, similar to the binding area of ssRNA-16 described above. Therefore, we speculate that suramin can competitively inhibit RNA binding to SARS-CoV-2 N-NTD by occupying the same binding sites with a binding affinity higher than RNA.

### 2.4. Structural Model of the SARS-CoV-2 N-NTD-Suramin Complex

To clarify the structural basis of SARS-CoV-2 N-NTD binding to suramin, we built a docking model of the SARS-CoV-2 N-NTD-suramin complex by using the HADDOCK 2.4 online server. The 3D structure of the N-NTD protein derived from that of the SARS-CoV-2 N-NTD-RNA complex (PDB ID: 7ACT) was used as the initial structural model for docking N-NTD with suramin. The model with the lowest interface energy was selected as the optimal docking model (hereinafter referred to as the docking model), which illustrates that suramin is captured in the positively charged cavity of SARS-CoV-2 N-NTD formed by the finger and palm subdomains (Figure 5a).

Then, we analyzed the structural model of N-NTD-suramin using the PLIP online tool. It is shown that the N-NTD protein binds suramin mainly through three salt bridges. In detail, R95 and K102 form salt bridges with sulfonic acid groups at both ends of suramin on one site, while on the other side, the guanidyl group of R88 forms a salt bridge with suramin. Furthermore, R92, R93, I94 and G96 form hydrogen bonds with suramin to further stabilize the N-NTD-suramin complex. Besides, I94 and A156 of N-NTD separately form two hydrophobic contacts with suramin (Figure 5b).

To identify the crucial residues for SARS-CoV-2 N-NTD capturing suramin, we constructed seven mutants (R88E, R92E, R93E, I94G, R95E, K102E and A156G), and determined their affinities for binding suramin by conducting BLI experiments. Compared with the wild-type N-NTD (WT), these mutants decreased the binding affinities of suramin by around 50% (Figure 6). In particular, both the R92E and K102E mutants significantly weakened the binding of N-NTD to suramin with a 59% reduction in affinity compared to the WT. The affinities of the R88E, R93E, R95E and A156G mutants to suramin were also decreased by about 51–52%. Besides, the I94G mutant slightly decreased the affinity to suramin by around 43% compared with the WT. These results suggest that residues R88, R92, R93, I94, R95, K102 and A156 play crucial roles in the interaction of N-NTD with suramin. Although we have not obtained the docking model of the N-NTD-ssRNA-16 complex due to the higher flexibility of ssRNA-16, it has been previously reported that the positively charged cavity between the palm and finger subdomains of SARS-CoV-2 N-NTD is the main RNA-binding area, and R92, R93, R95, K102 are four important residues for capturing RNA [14,18]. Future work should be performed to experimentally determine the 3D structure of the SARS-CoV-2 N-NTD-suramin complex.

### 2.5. Suramin Binding Changes the Dynamics Property of SARS-CoV-2 N-NTD

Protein dynamics characteristics are usually changed due to the binding of small molecules. We herein compared the dynamic properties of suramin-bound N-NTD with those of free N-NTD, to evaluate the effect of suramin binding on the N-NTD dynamics. We measured the longitudinal relaxation rate R_1_, transverse relaxation rate R_2_ and heteronuclear {^1^H}-^15^N hNOE data of free N-NTD and suramin-bound N-NTD (Figure 7). Except for 11 proline residues in the SARS-CoV-2 N-NTD and 8 residues with either invisible resonances or significant resonance overlap in the 2D ^1^H-^15^N NMR spectra, 118 residues were used for the backbone dynamics analysis of free N-NTD. As suramin binding induced significant peak broadening and even disappearance in the NMR spectra, only 98 residues were finally used for the backbone dynamics analysis of suramin-bound N-NTD.

The R_1_ distribution of free N-NTD ranged from 0.661 to 1.260 s^−1^, with an average value of 0.849 s^−1^. The R_1_ values of residues located in the loop regions between the N-terminus, C-terminus and β2-β3 segment were slightly higher than the average value, indicating that these regions were flexible. On the other hand, the R_1_ distribution of suramin-bound N-NTD ranged from 0.332 to 1.293 s^−1^, with an average range of 0.555 s^−1^ which is obviously lower than that of free N-NTD. The lowered R_1_ value suggested that the fast motion of the suramin-bound N-NTD on the ps-ns timescale was suppressed. However, the residues located at the loop region between β2 and β3 had higher R_1_ values at around 0.710 s^−1^ than the average value, implying that this segment still experienced faster motion on the ps-ns timescale after N-NTD binding to suramin. The averaged R_2_ value of the suramin-bound N-NTD was about 34.1 s^−1^, which was apparently higher than that of free N-NTD (14.3 s^−1^), indicating that the internal motion of N-NTD on μs-ms timescale was suppressed upon suramin binding. The hNOE values of free N-NTD were distributed between 0.067 and 0.938 with an average value of 0.770, and the residues on the loop region between β2 and β3 were significantly lower than the average value, indicating that this loop region was highly flexible in solution. The suramin-bound N-NTD basically had similar hNOE values as free N-NTD (Figure 7). These results indicated that both the ps-ns fast motion and μs-ms slow motion of residues in N-NTD are mostly suppressed after suramin binding. Notably, the loop between β2 and β3 still exhibited ps-ns fast motion, which might be related to the association and dissociation of suramin with N-NTD. As previously reported, conformational changes in the loop region between β2 and β3 can facilitate N-NTD binding RNA [19]. Therefore, it can be expected that the internal motion of the loop between β2 and β3 might facilitate N-NTD binding suramin.

We further processed the relaxation parameters by using Mathematica Notebooks to obtain low-spectral density functions J(0), medium-spectral density functions J(ω_N_) and high-spectral density functions J(0.87ω_H_) for free N-NTD and suramin-bound N-NTD (Figure 8). In free N-NTD, the locally averaged J(0) values of residues on the loop region between the N-terminus, C-terminus and β2-β3 were significantly lower than the globally averaged J(0) value of (5.3 × 10^−6^ s·rad^−1^). On the other hand, the locally averaged J(ω_N_) and J(0.87ω_H_) values of these loops were significantly higher than the globally averaged values of N-NTD (J(ω_N_), 2.1 × 10^−7^ s·rad^−1^; J(0.87ω_H_), 3.3 × 10^−9^ s·rad^−1^). These results indicated that these loop regions are structurally flexible with fast motion on the ps-ns timescale. The globally averaged J(0) value of suramin-bound N-NTD was 4.5 × 10^−6^ s·rad^−1^, which was obviously lower than that of free N-NTD, implying that the internal mobility on μs-ms timescale was somewhat suppressed. The average J(ω_N_) value of the suramin-bound N-NTD was 1.3 × 10^−7^ s·rad^−1^, which was also lower than that of free N-NTD, suggesting that suramin binding observably suppressed the structural flexibility of N-NTD on the ps-ns timescale. However, residues R93, K100 and D103 located at the loop between β2 and β3 still had higher J(ω_N_) values, implying that this loop kept certain flexibility in the ps-ns timescale. Although the average J(0.87ω_H_) value of the suramin-bound N-NTD was slightly different to that of free N-NTD, residues R95, D98 and D103 located at the loop between β2 and β3 still displayed high J(0.87ω_H_) values, suggesting that the ps-ns fast motion of this loop remains in the suramin-bound N-NTD. These results indicated that suramin binding makes the N-NTD protein more rigid and that the loop between β2 and β3 still exists in fast motion on the ps-ns timescale to facilitate the association and dissociation of suramin with N-NTD.

## 3. Materials and Methods

### 3.1. Cloning, Expression and Purification

The plasmid pET28a harboring the DNA fragment of SARS-CoV-2 N protein (pET28a-N) was provided by the Guangdong Laboratory Animals Monitoring Institute. Then, the DNA fragment corresponding to the N-terminal domain of SARS-CoV-2 N protein (N-NTD, residues 44–180) was amplified using PCR with pET28a-N as the template and inserted into the pSUMO vector containing an N-terminal 6xHis-tag followed by a SUMO fusion partner with a SUMO protease cutting site between them. Seven N-NTD mutants were generated using site-directed mutagenesis, including R88E, R92E, R93E, I94G, R95E, K102E and A156G. All recombinant plasmids used in this study were verified via DNA sequencing and transformed into *E. coli* BL21 (DE3) strain. The recombinant cells were then induced with 0.5 mM IPTG at OD_600_ nm of 0.6 and continually cultured at 25 °C for 12 h in LB liquid media. For preparing uniformly ^15^N-labeled N-NTD samples, the cells were cultured in M9 media with 0.1% (*m*/*v*) of ^15^NH_4_Cl as a nitrogen source.

The harvested cell pellets of SARS-CoV-2 N-NTD or its mutants were resuspended in 50 mM Tris, pH 8.0, 200 mM NaCl, 1.0 mM phenylmethylsulfonyl fluoride (PMSF), and lysed on ice using sonication. The soluble fraction of the lysate was collected using centrifugation and loaded onto 5 mL of Ni-NTA resin. After washing impurities, the target proteins were eluted with 50 mM Tris, pH 8.0, 200 mM NaCl, 250 mM imidazole. Thereafter, the protein was buffer-exchanged into an NMR buffer (25 mM Na_3_PO_4_, 50 mM NaCl, pH 6.5) and further purified through size exclusion chromatography (SEC) using ÄKTA FPLC system with a Superdex 75 10/300 GL column (GE Healthcare, Chicago, IL, USA) setting flow rate as 0.6 mL/min and alarm pressure as 1.5 MPa. Additionally, the purified proteins were incubated with 0.4 mg/mL SUMO-protease at room temperature for 3 h to remove the SUMO-tag, and the enzymatic mixture was further purified using second Ni-NTA affinity chromatography. Finally, the N-NTD target protein was eluted using the NMR buffer successively with 40 mM and 60 mM imidazole.

### 3.2. Gel Mobility Shift Assay

Binding reaction mixtures containing 25 μM ssRNA-16 (5′-AUAUGGAAGAGCCCUA-3′) and 25 μM N-NTD in 30 μL of NMR buffer were incubated at 25 °C for 30 min. Then, suramin (Sigma, St. Louis, MO, USA, product No. S2671-100MG) was gradually added into the reaction mixtures step-by-step in molar ratios of 1:1:1, 1:1:2.5, 1:1:5, 1:1:7.5, 1:1:10, 1:1:15 and 1:1:20. All samples were analyzed using electrophoresis on a 20% native polyacrylamide gel and stained with 50 mL of dye solution containing 50 mL of H_2_O, 0.292 g NaCl and 15 μL of 4S Red Plus Nucleic.

### 3.3. BLI Assays

BLI experiments were performed at 298 K on ForteBio OCTET96 to measure the affinities of SARS-CoV-2 N-NTD or its mutants for binding ligand (ssRNA-16 or suramin). All recombinant proteins were dissolved in 25 mM Na_3_PO_4_, 50 mM NaCl, pH 6.5. Firstly, 20 μM biotinylated N-NTD protein was loaded onto the super streptomycin probe, and ssRNA-16/suramin at a series of concentrations (0.62, 1.25, 2.5, 5.0, 10 and 20 μM) were gradually associated with recombinant proteins. Then, the dissociation was processed in the buffer of 25 mM Na_3_PO_4_, 50 mM NaCl, pH 6.5. The probes without biotinylated protein were used as blank. The association time and dissociation time for N-NTD interacting with ligands were set to be 120 s and 200 s, respectively. The loading time and baseline time were set to be 300 s and 200 s, respectively. ForteBio Data Analysis 9.0 software was used to obtain the dissociation constants of N-NTD proteins and ligands.

### 3.4. NMR Titration Assays

2D ^1^H-^15^N HSQC spectra were recorded at 298K on a Bruker Avance III 850 MHz spectrometer equipped with a ^1^H/^13^C/^15^N TCI cryogenic probe. All protein samples for NMR spectroscopy were dissolved in the NMR buffer containing 5% D_2_O for magnetic field lock. All spectra were processed with NMRPipe and analyzed with NMRFAM-SPARKY. For suramin titration, suramin was added to 50 μM N-NTD protein solution in molar ratios of 1:0.5, 1:1, 1:1.5, 1:2 and 1:2.5. For ssRNA-16 titration, the ssRNA-16 (5′-AUAUGGAAGAGCCCUA-3′) was added to 50 μM N-NTD protein solution at molar ratios of 1:1, 1:2, 1:3 and 1:4. 2D ^1^H-^15^N HSQC spectra were recorded at each titration point at 298K. The chemical shift perturbation (CSP) was determined with the empirical formula:(1)$$\Delta \delta =\sqrt{\frac{1}{2}\left[{\left(\Delta \delta_{H}\right)}^{2}+0.14{\left(\Delta \delta_{N}\right)}^{2}\right]},$$in which Δδ_H_ and Δδ_N_ represented the chemical shift displacements of N-NTD for ^1^H and ^15^N nuclei observed upon ligand titration, respectively.

The intensity perturbation index (IPI) was determined with the empirical formula:(2)$$\mathrm{IPI}=1\;-\;H\left(\mathrm{complex}\right)/H\left(\mathrm{free}\right),$$in which H(complex) and H(free) represented peak intensities for ligand-bound N-NTD and free N-NTD.

### 3.5. Molecular Docking

The HADDOCK webserver (https://wenmr.science.uu.nl/haddock2.4/ (accessed on 8 March 2022)) was employed to build the structural model of the N-NTD-suramin complex. The 3D structure of N-NTD was obtained from Protein Data Bank (https://www.rcsb.org/ (accessed on 8 March 2022)) (PDB ID: 7ACT). The docking sites were defined based on the coordinate centers of residues with peaks undergoing significant chemical shift perturbations or peak broadening in 2D ^1^H-^15^N HSQC spectra. Set the number of rigid-body structures for docking to 10,000, and all the numbers of semi-flexible refinements, final refinements and analyses to 1000. The lowest-energy conformation model with the highest score was analyzed by the PLIP webserver (https://plip-tool.biotec.tu-dresden.de/plip-web/plip/index (accessed on 8 March 2022)). The structural representations were prepared with the PyMOL program.

### 3.6. NMR Relaxation Measurements

The N-NTD protein sample at a concentration of 1.2 mM was used to conduct NMR relaxation measurements of R_1_, R_2_ and hNOE. R_1_ values were calculated with relaxation delays of 10, 50, 100 (×2), 200, 400, 600, 800 (×2), 1200, 1600 and 2000 ms, while R_2_ values were determined with relaxation delays of 16.32, 32.64 (×2), 48.96, 65.28, 81.60, 97.92, 114.24, 130.56 (×2), 146.88, and 163.20 ms. The hNOE values were obtained in interleaved spectra with and without a 3 s ^1^H pre-saturation, the latter being replaced by a 3 s relaxation delay. NMRFAM-SPARKY was used to fit exponential decay curves to the experimental serial data for determining R_1_ and R_2_ rates. The relaxation measurement for suramin-bound N-NTD was also performed following the same approach.

### 3.7. Reduced Spectral Density Mapping

The relaxation rates R_1_, R_2_ and hNOE were used to map the spectral density parameters J(0), J(ω_N_) and J(0.87ω_H_) following the approach of spectral density function [31]. The calculations were implemented using the script reported by Leo Spyracopoulos [32].

## 4. Conclusions

As one of the highly conserved proteins in coronaviruses, the N protein binds the viral RNA and packs it into the ribonucleoprotein (RNP) complex. Therefore, the N protein acts as a promising target for developing drugs against coronaviruses. Herein, we found suramin can disturb the association of the NTD domain of SARS-CoV-2 N protein (SARS-CoV-2 N-NTD) with RNA through competitive binding to the RNA-binding area on the protein. Compared with RNA, suramin possesses a significantly higher affinity for binding to SARS-CoV-2 N-NTD. Residues R88, R92, R93, I94, R95, K102 and A156 play crucial roles in SARS-CoV-2 N-NTD capturing suramin. Furthermore, suramin binding increases the structural rigidity of the N-NTD protein by suppressing global ps-ns fast motion and μs-ms slow motion. However, the ps-ns fast motion of the loop between β2 and β3 remains, potentially facilitating the association and dissociation of suramin with N-NTD. Our results not only disclose the structural basis for suramin disturbing the interaction between SARS-CoV-2 N-NTD and RNA but also shed light on the development of drugs against SARS-CoV-2 and other coronaviruses.

## Figures and Tables

**Figure 1 molecules-28-02534-f001:**
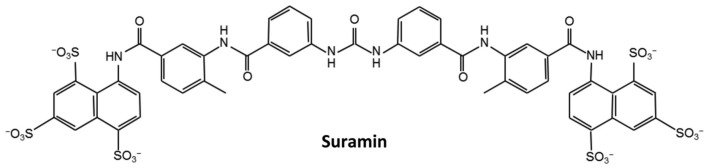
The chemical structure of suramin.

**Figure 2 molecules-28-02534-f002:**
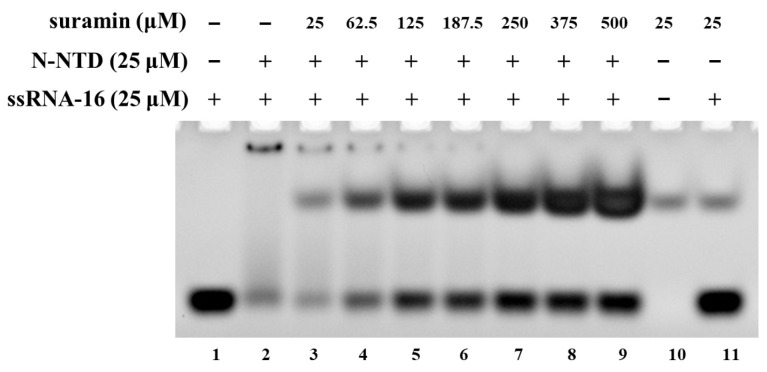
EMSA experiment for suramin disturbing the association between SARS-CoV-2 N-NTD and ssRNA-16. Lane 1: free ssRNA-16; lane 2: mixture of N-NTD with ssRNA-16 at the molar ratio of 1:1; lane 3–9: mixture of N-NTD with ssRNA-16 and suramin at molar ratios of 1:1:1, 1:1:2.5, 1:1:5, 1:1:7.5, 1:1:10, 1:1:15, 1:1:20; lane 10: suramin; lane 11: mixture of suramin with ssRNA-16 at the ratio of 1:1.

**Figure 3 molecules-28-02534-f003:**
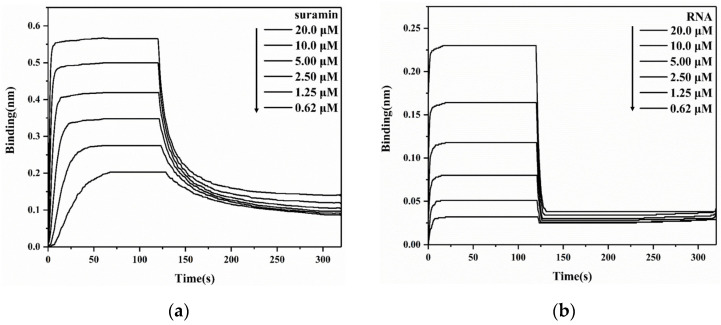
BLI experiments for detecting the interaction of SARS-CoV-2 N-NTD with suramin (**a**) or ssRNA-16 (**b**).

**Figure 4 molecules-28-02534-f004:**
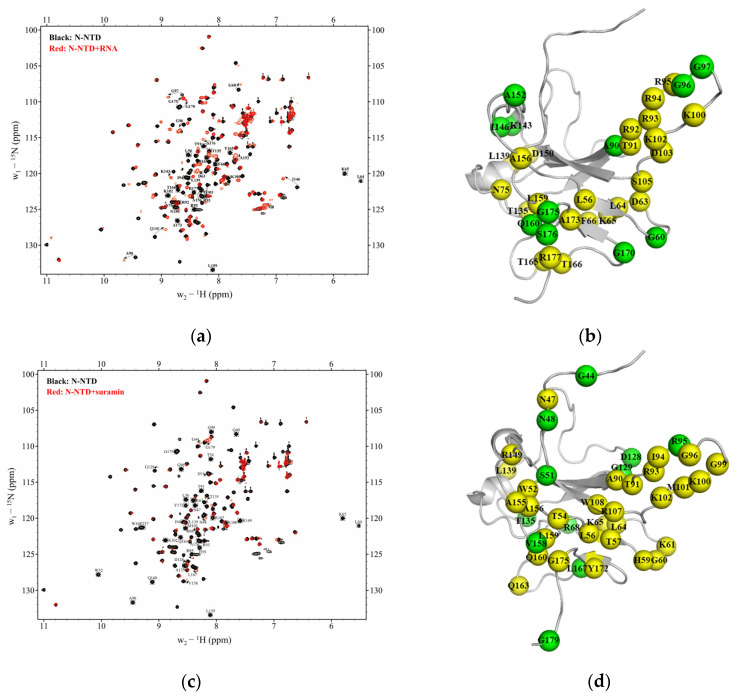
Binding site comparison of ssRNA-16 and suramin on SARS-CoV-2 N-NTD. (**a**) 2D ^1^H-^15^N HSQC spectra of N-NTD titrated with ssRNA-16. (**b**) Mapping the residues perturbed during the RNA titration on the 3D structure of N-NTD (PDB ID: 6YI3). (**c**) 2D ^1^H-^15^N HSQC spectra of N-NTD titrated with suramin. (**d**) Mapping the residues perturbed during the suramin titration on the 3D structure of N-NTD (PDB ID: 6YI3). The broad peaks are indicated in the dashed rectangles in the spectra and the corresponding residues are shown in yellow in the structure, while the shifted peaks are indicated as arrows and the corresponding residues are shown in green.

**Figure 5 molecules-28-02534-f005:**
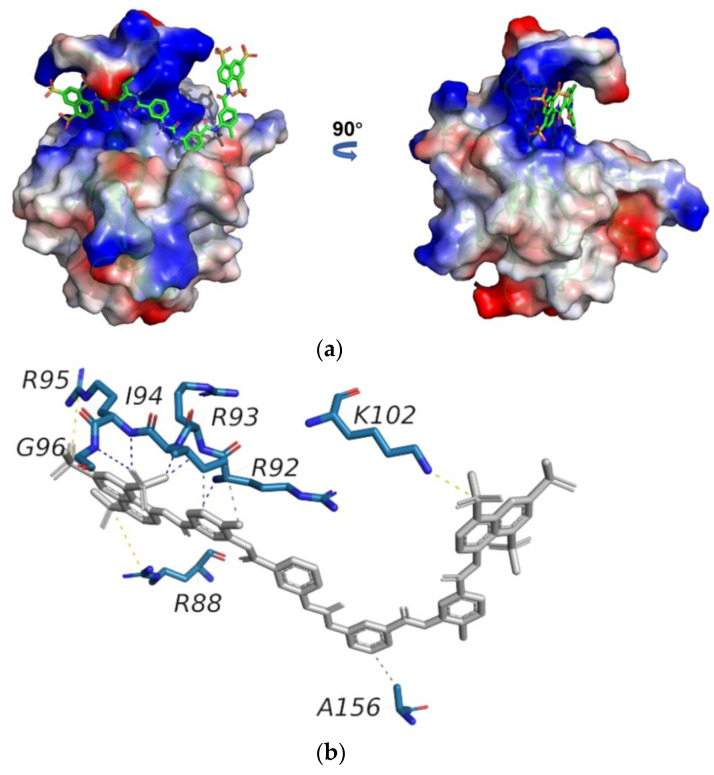
Structural model of the SARS-CoV-2 N-NTD-suramin complex. (**a**) The docking model was built using HADDOCK. Suramin is shown in grey. (**b**) Schematic diagram of intermolecular interactions in the structural model of N-NTD-suramin produced using PLIP 2.2.0 online software.

**Figure 6 molecules-28-02534-f006:**
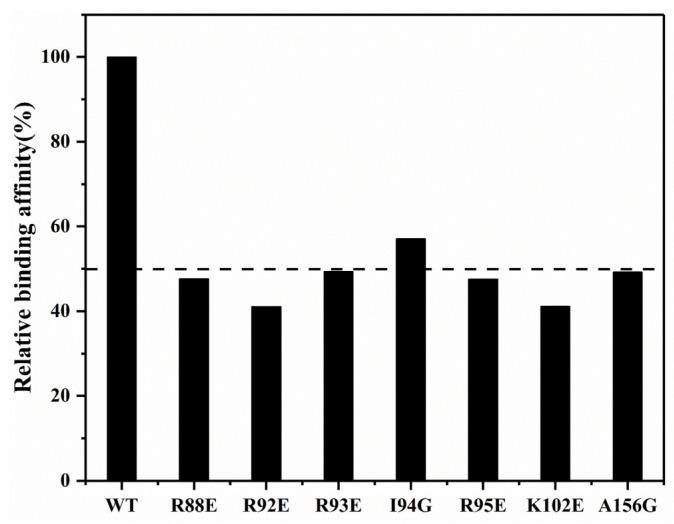
Affinity comparison of SARS-CoV-2 N-NTD WT and mutants for binding suramin. The dashed line denotes the relative binding affinity of 50%.

**Figure 7 molecules-28-02534-f007:**
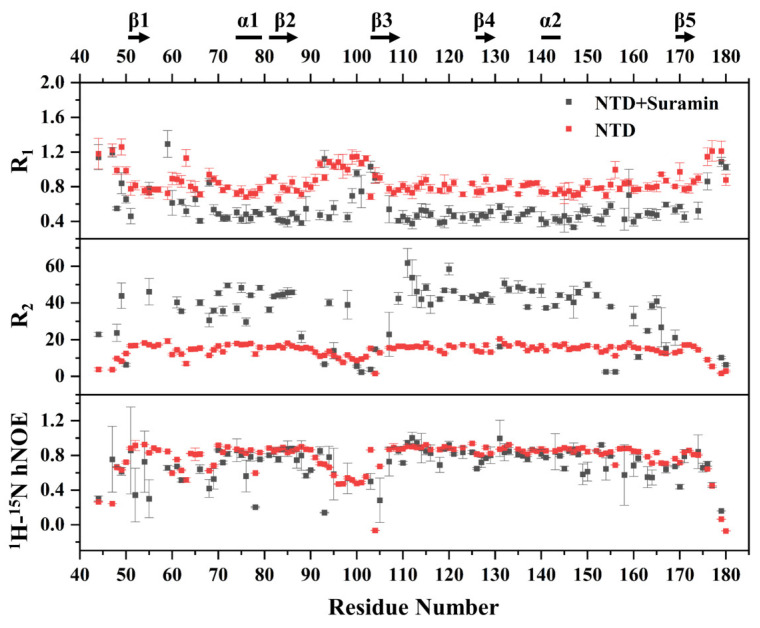
NMR-measured backbone relaxation parameters R_1_, R_2_ and hNOE values of free N-NTD and suramin-bound N-NTD.

**Figure 8 molecules-28-02534-f008:**
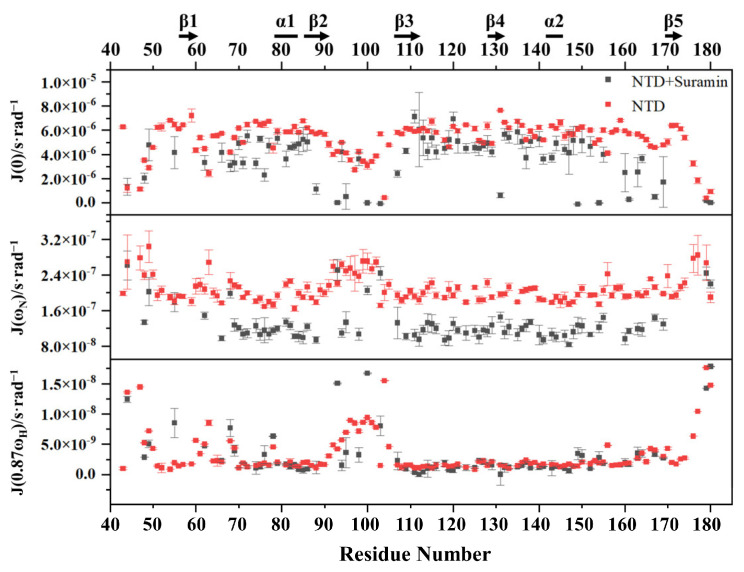
Calculated spectral density parameters of free N-NTD and suramin-bound N-NTD.

## Data Availability

The data presented in this study are available upon request from the corresponding author.

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
