# Peer review of "Suramin Disturbs the Association of the N-Terminal Domain of SARS-CoV-2 Nucleocapsid Protein with RNA"

_molecules, 2023, doi:10.3390/molecules28062534_

Round 1

Reviewer 1 Report

The manuscript presented by Guo et al. describes that Suramin can be a drug candidate due to its ability to inhibit the interactions between NTDs of SARS-CoV-2  nucleocapsid protein and RNA. The manuscript is well written and scientifically sound. However, I have few concerns for authors which will be provided in the authors sections to make the manuscript better. I endorse the publication of this manuscript. 

1. Did Authors check the potency of Suramin in preventing SARS-CoV-2 infection? Any details would be interesting in this regard.

2. What is Suramin and what is its source? Can you provide a structure of this molecule that will be helpful for readers?

3. Is Suramin only specific for SARS-CoV-2 or can be used against other coronaviruses?

4. Which part of SARS-CoV-2 genome did you take the 16-mer RNA sequence and why only this sequence combination? Provide a reasoning.

5. In line, 104-107. How do you know the residue positions? Do you have a three dimensional structure? If yes, then why didn't you consider determining the structure of the NTDs and Suramin complex by NMR?

Author Response

Dear reviewer,

We would like to thank you for valuable comments on our manuscript titled "Suramin disturbs the association of the N-terminal domain of SARS-CoV-2 nucleocapsid protein with RNA" (ID molecules-2245748). Following these comments, we have endeavored to carefully revised our manuscript. For your convenience, these revisions are highlighted in red in the revised manuscript. The detailed point-by-point responses to the reviewer’s comments are attached below.

Professor Donghai Lin

College of Chemistry and Chemical Engineering,

Xiamen University, Xiamen 361005.

Reviewer 2 Report

The work exerted is so appreciated. A nice idea was discussed here. I was really impressed by it. Some little points need to be addressed before acceptance. So, a minor revision may be required to improve the manuscript.

1-      A reference drug is highly recommended (if Possible) to be included in the conducted in vitro and in silico investigations.

2-      The manuscript should include a separate section for “Conclusion”.

Author Response

(The authors gave the same response as above.)
